# The Impact of Maladaptive Coping Styles on Psychological Outcomes in Tuberculosis Patients

**DOI:** 10.3390/healthcare13091042

**Published:** 2025-05-01

**Authors:** Ion Papava, Ana-Maria Cristina Daescu, Liana Dehelean, Ana-Cristina Bredicean, Adrian Cosmin Ilie, Sorin Ursoniu, Mariana Bondrescu, Ion Radu, Andrei Daescu, Alexandru-Ioan Gaitoane, Cristian Oancea

**Affiliations:** 1Neurosciences Department, Discipline of Psychiatry, “Victor Babes” University of Medicine and Pharmacy, 300041 Timisoara, Romanialianadeh@umft.ro (L.D.); bredicean.ana@umft.ro (A.-C.B.); mariana.bondrescu@umft.ro (M.B.); 2Center for Cognitive Research in Neuropsychiatric Pathology (NeuroPsy-Cog), “Victor Babes” University of Medicine and Pharmacy, 300041 Timisoara, Romania; 3Department of Psychiatry, Timis County Emergency Clinical Hospital “Pius Brinzeu”, 300041 Timisoara, Romania; daescu.andrei@hosptm.ro (A.D.); gaitoane.alexandru-ioan@hosptm.ro (A.-I.G.); 4Psychiatry Compartment, “Dr. Victor Popescu” Emergency Military Clinical Hospital, 300080 Timisoara, Romania; 5Department III Functional Sciences, Division of Public Health and Management, “Victor Babes” University of Medicine and Pharmacy, 300041 Timisoara, Romania; ilie.adrian@umft.ro (A.C.I.); sursoniu@umft.ro (S.U.); radu.ion@umft.ro (I.R.); 6Center for Translational Research and Systems Medicine, “Victor Babes” University of Medicine and Pharmacy, 300041 Timisoara, Romania; 7Center for Research and Innovation in Personalized Medicine of Respiratory Diseases (CRIPMRD), “Victor Babes” University of Medicine and Pharmacy, 300041 Timisoara, Romania; oancea@umft.ro

**Keywords:** TB, COPE questionnaire, GAD7, PHQ9

## Abstract

Background/Objective: Tuberculosis (TB) is associated with significant psychological distress, including anxiety and depression, which may be influenced by coping styles. This study aimed to evaluate the relationship between coping mechanisms, psychological outcomes, and sociodemographic factors in TB patients. Methods: A total of 100 TB patients admitted to the Victor Babeș Clinical Hospital of Infectious Diseases and Pneumophtisiology, Timișoara, were assessed using the COPE questionnaire for coping styles and the GAD7 and PHQ9 scales for anxiety and depression. The Wilcoxon signed-rank test analyzed the changes in the psychological scores between admission and discharge. Results: Multinomial and linear regression analyses identified the predictors of coping styles based on psychological and sociodemographic factors. Anxiety and depression significantly improved during hospitalization (PHQ9: *p* < 0.001, GAD7: *p* < 0.001). Social-support-focused coping showed the largest depression reduction (PHQ9: from 13 to 4), while avoidant coping had the lowest residual distress (PHQ9 = 0.5, GAD7 = 0). Age and marital status were significant predictors of problem-focused coping, with older and married patients being more likely to adopt this strategy (β = 0.08, *p* = 0.008). Coping styles significantly influence psychological outcomes in TB patients. Problem-focused coping was associated with better psychological recovery, while social-support-focused coping was linked to persistent distress. Conclusions: Integrating mental health screening into TB care and tailoring interventions to coping styles may enhance psychological resilience and potentially support treatment adherence, a relationship that should be further explored in future research.

## 1. Introduction

Tuberculosis (TB) remains a critical global health issue, particularly in low- and middle-income countries, where healthcare access is limited and socio-economic barriers exacerbate the disease burden [1,2,3]. While advances in diagnosis and treatment have improved clinical outcomes, the psychological impact of TB is often overlooked.

TB patients frequently face stigma, isolation, and financial stress, which—alongside the prolonged treatment duration and side effects of anti-TB drugs—contribute to high rates of anxiety and depression. These mental health challenges can negatively influence treatment adherence and clinical outcomes, including the risk of developing drug-resistant strains [4,5]. Estimates show that 40–70% of TB patients report symptoms of anxiety or depression, particularly in settings where stigma and limited psychosocial support are prevalent [6,7,8,9].

Coping strategies—defined as cognitive and behavioral efforts to manage stress—play a central role in shaping psychological outcomes. These are typically categorized as problem-focused, emotion-focused, avoidant, and social-support-focused coping [10,11,12,13]. While problem-focused strategies are linked to better outcomes, avoidant coping has been associated with increased distress. Social support may buffer stress, but its effectiveness varies depending on the cultural context and the quality of support networks.

Patients with TB may adopt maladaptive coping mechanisms, particularly when confronting chronicity, stigma, or limited resources. Studies show that psychological distress tends to decrease during hospitalization, but persistent maladaptive coping styles can delay recovery [14,15].

Sociodemographic factors influence coping patterns. For example, younger patients may favor avoidant coping, while older individuals more often engage in problem- or emotion-focused strategies. Gender, education level, and socio-economic status further modulate coping behaviors [16,17]. Cultural background also shapes coping preferences: collectivist societies may encourage social-support-focused coping, while individualistic cultures may favor problem-solving or avoidance [2,18,19,20].

Integrating mental health support into TB care is essential, particularly in resource-limited settings. Approaches such as involving community health workers or using digital tools (e.g., telepsychiatry) show promise in addressing psychological needs [21,22,23]. However, further research is required to understand how coping strategies evolve over time and how they relate to psychological outcomes.

This study aims to examine the relationship between coping styles, psychological distress (anxiety and depression), and sociodemographic variables in TB patients. By assessing the changes in mental health during hospitalization, our findings aim to inform the development of personalized, culturally appropriate psychosocial interventions.

## 2. Materials and Methods

### 2.1. Study Design and Participant Demographics

This study included a cohort of 100 patients newly diagnosed with active pulmonary tuberculosis, who were admitted to Victor Babeș Clinical Hospital for Infectious Diseases and Pneumophtisiology, Timișoara, specifically to Pulmonology Clinic I. Data collection was conducted prospectively over the course of one year, starting in January 2024, and all the patients were evaluated during their hospital stay. Sociodemographic information, including age, sex, marital status, residence (urban or rural), smoking status, and educational level, was collected at admission through structured interviews and medical records. Psychological assessments were performed using validated tools to measure anxiety and depression levels: the Generalized Anxiety Disorder-7 (GAD7) scale for anxiety and the Patient Health Questionnaire-9 (PHQ9) for depression. These scales were administered to all the patients at two time points: upon admission (baseline), prior to the initiation of anti-tuberculosis treatment, and at discharge. Coping styles were assessed through the Coping Orientation to Problems Experienced (COPE) questionnaire, which evaluates four coping strategies: avoidant coping, emotion-focused coping, problem-focused coping, and social-support-focused coping. Each patient was assigned to one coping style based on their highest individual score. The data collection process adhered to ethical standards, ensuring patient confidentiality and informed consent. This comprehensive dataset provided a foundation for analyzing the relationships between coping styles, psychological outcomes, and sociodemographic factors in patients with tuberculosis.

### 2.2. Inclusion and Exclusion Criteria

The patients included in this study were those with a confirmed diagnosis of pulmonary tuberculosis, determined based on clinical, radiological, or microbiological criteria. All the patients were required to be 18 years or older, capable of providing informed consent and fully understanding the study procedures, ensuring reliable responses to the psychological assessments. Participant understanding was assessed through structured interviews conducted by trained clinicians, who explained this study using simplified language and verified comprehension using open-ended questions.

Patients were excluded if they had pre-existing psychiatric conditions, such as major depressive disorder or generalized anxiety disorder, or if they were receiving ongoing treatment with psychotropic medication, including antidepressants, antipsychotics, or anxiolytics, as these factors could confound the psychological assessments and coping style evaluations. Patients who failed to complete any of the required assessments, including the GAD7, PHQ9, or COPE questionnaire, were excluded. Participants with severe comorbidities, such as advanced cancer or end-stage organ failure, were excluded, as these conditions could independently influence the coping mechanisms and psychological outcomes. Patients with multidrug-resistant tuberculosis (MDR-TB) were also excluded due to the unique challenges and psychological burden associated with this condition. Pregnant patients were not included to avoid the potential confounding effects of pregnancy on the psychological and coping evaluations. Additionally, individuals with active substance use disorders or cognitive impairments that could interfere with the accurate completion of the GAD7, PHQ9, and COPE questionnaire were excluded.

### 2.3. Psychometric Assessments

The COPE Questionnaire, developed by Carver et al. (1989), is a psychological instrument designed to evaluate the coping mechanisms individuals use in response to stress. It incorporates the theoretical framework of Lazarus and Folkman (1987) and categorizes coping strategies into four main groups: problem-focused coping (e.g., active coping, planning, suppression of competing activities), emotion-focused coping (e.g., positive reinterpretation, restraint, acceptance, religious coping), social-support-focused coping (e.g., seeking instrumental or emotional support, emotional expression), and avoidant coping (e.g., denial, mental disengagement, behavioral disengagement, substance use, humor). The Romanian adaptation of the COPE Questionnaire by Crașovan and Sava includes 60 items grouped into 15 distinct coping strategies, with each strategy assessed using 4 items rated on a Likert scale from 1 (“I don’t do this at all”) to 4 (“I do this a lot”). The total score for each strategy ranges from 4 to 16, with higher scores indicating greater use of that coping mechanism. To determine an individual’s dominant coping style, the scores from related strategies within each category are summed, and the category with the highest cumulative score is identified. The instrument has demonstrated good internal consistency in the Romanian population, with the Cronbach’s alpha values ranging from 0.48 to 0.92 across the subscales and an average of 0.70, making it a valuable tool for both research and clinical applications [24,25,26].

The PHQ-9 (Patient Health Questionnaire-9) is a validated self-administered questionnaire designed to assess the severity of depressive symptoms. It consists of 9 items, each rated on a scale from 0 (not at all) to 3 (nearly every day), with a total score ranging from 0 to 27. The depression severity is classified into five categories: minimal or no depression (0–4), mild depression (5–9), moderate depression (10–14), moderately severe depression (15–19), and severe depression (20–27). The PHQ-9 has been shown to have excellent internal consistency, with a Cronbach’s alpha of 0.897 in the Romanian population, and is widely recognized for its reliability and validity. It is a quick and effective tool for screening and monitoring depression, making it valuable in both clinical and research settings [27].

The GAD-7 (Generalized Anxiety Disorder-7) is a brief, self-administered questionnaire widely used to screen for and assess the severity of generalized anxiety disorder. It consists of 7 items rated on a scale from 0 (not at all) to 3 (nearly every day), yielding a total score ranging from 0 to 21. The anxiety severity is categorized as minimal (0–4), mild (5–9), moderate (10–14), or severe (15–21). The Romanian version of the GAD-7 has been validated in both clinical and non-clinical populations, demonstrating good psychometric properties, including strong internal consistency and convergent validity. The scale strongly correlates with the Depression, Anxiety, and Stress Scale-21 and moderately correlates with the State-Trait Anxiety Inventory, confirming its reliability and validity across diverse settings. The GAD-7 is an effective and efficient tool for identifying and monitoring anxiety, with broad applicability in both research and clinical practice [28].

### 2.4. Statistical Analysis

In the statistical analysis, the variables were presented based on their distribution characteristics. Numerical variables were assessed for normality using the Shapiro–Wilk test. For variables with non-Gaussian distributions (*p* < 0.05), non-parametric approaches were employed throughout the analysis. The descriptive statistics for these variables were reported as medians with interquartile ranges (Q25–Q75). For the numerical variables with *p* > 0.05 in the Shapiro–Wilk test, the data were described using means and standard deviations (Mean ± SD). For the categorical variables, counts and proportions were used to summarize the data. In the statistical analysis, the differences between coping styles and numerical variables, such as age or the GAD7 and PHQ9 scores, were evaluated using the Kruskal–Wallis test due to the non-Gaussian distribution of these variables. For the differences between coping styles and the categorical variables, such as sex or marital status, Pearson’s Chi-squared test was applied. To analyze the differences between each specific type of coping and the GAD7 and PHQ9 scores, both at admission and at discharge, the Wilcoxon signed-rank test was employed, as these measurements were repeated for the same patients. For the correlations between each type of coping and the GAD7 and PHQ9 scores at admission and discharge, the Spearman rank correlation was used to assess the strength and direction of the relationships. Additionally, multinomial regression and linear regression models were developed to investigate the associations between coping styles, sociodemographic variables, and anxiety and depression scores. Multinomial regression models were developed to investigate the associations between coping styles (as response variables), sociodemographic factors, and GAD7 and PHQ9 scores. The models employed the backward elimination method to optimize the predictor selection, with the Akaike information criterion (AIC) and Bayesian information criterion (BIC) used as criteria to select the most parsimonious models. The performance of the multinomial regression models was evaluated using Nagelkerke R^2^ to quantify the proportion of variance explained by the predictors. Linear regression models were constructed for each coping style to identify significant predictors among the sociodemographic variables and GAD7 and PHQ9 scores. The predictor selection followed the backward elimination method, with the AIC and BIC used to ensure optimal model fit. The performance of the linear regression models was assessed using the adjusted R^2^ to account for the number of predictors included. These models provided insights into the predictors of coping styles and their interactions with the other study variables. All the statistical analyses were conducted with a significance threshold of *p* < 0.05, and the results were reported with the 95% confidence intervals where applicable. The analyses were performed using R (version 4.3.0; R Core Team, 2023) and RStudio (version 2023.06.0 + 421; RStudio Team, 2023, Boston MA, USA), ensuring robust and reproducible findings.

## 3. Results

### 3.1. Descriptive Analysis of Cohort Characteristics and Clinical Outcomes

In this study, a detailed descriptive analysis was conducted to evaluate the relationships between coping styles, sociodemographic variables, and clinical outcomes, including the anxiety and depression levels. The analysis of the numerical and categorical variables offers critical insights into the cohort’s characteristics and coping mechanisms. The median age of the cohort was 43.50 years (Q25–Q75: 30.00–53.25), with a Shapiro–Wilk test *p*-value of <0.001, indicating a non-Gaussian distribution. The anxiety levels, assessed using the GAD7 scale, showed a median score of 7.00 (Q25–Q75: 4.00–12.00) at admission and decreased significantly to 2.00 (Q25–Q75: 1.00–5.00) at discharge, reflecting an improvement in symptoms. The depression levels, measured using the PHQ9 scale, followed a similar trend, with a median score of 9.00 (Q25–Q75: 4.00–13.00) at admission and decreasing to 3.00 (Q25–Q75: 2.00–5.00) at discharge. The *p*-values for these variables were all <0.001, confirming the non-Gaussian distributions. The coping styles also showed distinct patterns. Avoidant coping and social-support-focused coping were non-normally distributed, with median scores of 28.00 (Q25–Q75: 25.00–30.25) and 31.00 (Q25–Q75: 28.00–32.00), respectively. Conversely, problem-focused coping and emotion-focused coping demonstrated Gaussian distributions, with mean scores and standard deviations of 32.72 ± 4.08 and 34.7 ± 3.82, respectively, and Shapiro–Wilk *p*-values > 0.05. Data is presented in Table 1. 

The distribution of the categorical variables provides important insights into the sociodemographic and psychological characteristics of the cohort, all of whom were diagnosed with tuberculosis. A slight male predominance was observed, with 57% of participants being male and 43% female. The residence variable was almost evenly distributed, with 51% of participants residing in urban areas and 49% in rural areas, reflecting a balanced representation of different living environments. The smoking prevalence was notable, as 57% of participants were smokers compared to 43% who were non-smokers, suggesting a potentially high-risk population in terms of comorbidities. The educational attainment varied, with most participants (67%) having completed high school or less, while only 33% pursued higher education, which may influence health literacy and coping strategies. In terms of marital status, most participants were married (66%), while the remaining 34% were single, potentially reflecting differences in social support systems. Regarding the coping mechanisms, emotion-focused coping was the most prevalent strategy, utilized by 44% of participants, followed by problem-focused coping (31%) and social-support-focused coping (21%). Avoidant coping was the least common, employed by only 4% of participants, indicating limited reliance on this potentially maladaptive strategy. Data is presented in Table 2. 

### 3.2. Differences Among Coping Styles Based on Sociodemographic Variables

Table 3 presents the differences in coping styles across the categorical sociodemographic variables, including sex, residence, smoking status, educational level, and marital status. The coping styles—avoidant coping, emotion-focused coping, problem-focused coping, and social-support-focused coping—are reported as percentages within each subgroup, and the significance of the differences was assessed using the Chi-squared Pearson test. The distribution of coping styles varied across the subgroups. For example, emotion-focused coping was the most frequently reported strategy among both females (42%) and males (46%), whereas avoidant coping was minimally employed, with only 7% of females and 2% of males using this strategy. Residence status showed little variation, with emotion-focused coping being the dominant strategy for participants from both rural (41%) and urban (47%) areas. Smoking status revealed a borderline significant difference (*p* = 0.05), with smokers being more likely to employ social-support-focused coping (26%) compared to non-smokers (14%), who tended to favor problem-focused coping (42%). Regarding educational attainment, participants with a higher education level reported slightly higher use of social-support-focused coping (24%) compared to those with high school or less (19%), while emotion-focused coping remained the predominant strategy in both groups. Marital status also did not reveal significant differences (*p* = 0.56), with emotion-focused coping being the most common among married (48%) and single (42%) participants.

Table 4 presents the differences in coping styles across the numerical sociodemographic variable age, analyzed using the Kruskal–Wallis test. The median ages are reported for each coping style, along with the corresponding *p*-value, which evaluates the statistical significance of the observed differences. The analysis revealed some variation in the median ages among the coping styles. Participants employing problem-focused coping had the highest median age (48.00 years), suggesting this strategy may be more commonly used by older individuals. Conversely, those utilizing social-support-focused coping had the lowest median age (39.00 years), potentially indicating a preference for this strategy among younger individuals. Participants using avoidant coping and emotion-focused coping had identical median ages (41.50 years), representing a middle range within the cohort. The Kruskal–Wallis test yielded a *p*-value of 0.24, indicating that the differences in age across coping styles were not statistically significant.

### 3.3. Differences in Anxiety and Depression Scores at Admission and Discharge Across Coping Styles

Significant differences in the GAD7 and PHQ9 scores at admission and discharge were observed across the four coping styles, analyzed using the Kruskal–Wallis test. The scores are presented as the medians for each coping style, with the corresponding *p*-values indicating statistically significant differences. The analysis revealed significant differences in anxiety levels (GAD7) across the coping styles, both at admission (*p* < 0.01) and at discharge (*p* < 0.001). At admission, participants utilizing social-support-focused coping exhibited the highest GAD7 scores (median: 12.00), reflecting higher anxiety levels, while those using avoidant coping reported the lowest scores (median: 3.00). By discharge, the GAD7 scores decreased across all the coping styles, with the lowest anxiety levels observed in participants using avoidant coping (median: 0.00) and the highest in those relying on social-support-focused coping (median: 6.00). Similarly, significant differences were observed in the depression levels (PHQ9) at both admission and discharge (*p* < 0.01). At admission, participants employing social-support-focused coping had the highest median PHQ9 scores (13.00), while those utilizing avoidant coping exhibited the lowest scores (4.50). Following discharge, the PHQ9 scores showed improvements across all the coping styles, with the lowest scores again reported in the avoidant coping group (median: 0.50). These findings, summarized in Table 5, highlight the variability in anxiety and depression levels associated with the different coping styles. The results suggest that certain coping styles, such as avoidant coping, may be linked to lower anxiety and depression levels post-treatment, while others, such as social-support-focused coping, may correspond to persistently higher scores, reflecting differing psychological outcomes based on coping mechanisms.

### 3.4. Changes in Anxiety and Depression Scores Across Coping Styles

Figure 1 illustrates the differences in the PHQ9 scores at admission and discharge across the four coping styles, analyzed using the Wilcoxon signed-rank test. The boxplots represent the distribution of the depression levels for each coping style at the two points. Participants employing social-support-focused coping (n = 21) showed the most substantial improvement in the PHQ9 scores, with the median decreasing from 13 at admission to 4 at discharge—a threefold reduction. This suggests that individuals relying on social connections as a coping strategy may experience a notable alleviation of their depressive symptoms over time. For emotion-focused coping, the largest group in the cohort (n = 44), the median PHQ9 score decreased from 9 to 2, indicating a significant reduction in depressive symptoms. Similarly, participants using problem-focused coping (n = 31) experienced an improvement, with the median score decreasing from 6 at admission to 3 at discharge. In the avoidant coping group, the smallest subgroup (n = 4), the median PHQ9 score decreased from 4.5 to 0.5, suggesting minimal residual depressive symptoms at discharge. While all the coping styles were linked to reductions in the depression levels, the largest improvement in the median scores was observed in the social-support-focused coping group.

The statistical analysis using the Wilcoxon signed-rank test showed that the reductions in the PHQ-9 scores were statistically significant in the emotion-focused, problem-focused, and social-support-focused coping groups (*p*-adjusted < 0.001 for all three). In contrast, the reduction observed in the avoidant coping group did not reach statistical significance (*p*-adjusted = 0.5), most likely due to the small sample size (n = 4). The reported *p*-adjusted values reflect the corrections for multiple comparisons, applied to reduce the risk of Type I errors when testing multiple groups simultaneously. These findings suggest that while the depressive symptoms improved across all the coping styles, the observed changes were statistically robust only for the larger coping groups (Table 6).

Figure 2 illustrates the differences in the GAD7 scores at admission and discharge across the four coping styles, analyzed using the Wilcoxon signed-rank test. The boxplots show the distribution of the anxiety levels at the two points for each coping style, highlighting variations in the degree of improvement. Participants using social-support-focused coping (n = 21) showed the highest anxiety levels at admission, with a median GAD7 score of 12. By discharge, the median had decreased to 6, reflecting a 50% reduction in anxiety. This coping style, however, remained associated with the highest residual anxiety levels compared to the other strategies. For emotion-focused coping, the largest group in the cohort (n = 44), the median GAD7 score decreased from 7 to 2, indicating a significant reduction in anxiety symptoms over time. Participants using problem-focused coping (n = 31) experienced a decrease from a median score of 5 at admission to 4 at discharge, reflecting a smaller improvement. In the avoidant coping group, the smallest subgroup (n = 4), the median GAD7 score decreased from 3 at admission to 0 at discharge, suggesting a complete resolution of anxiety symptoms in this group. While all the coping styles were associated with reductions in the anxiety levels, social-support-focused coping showed the largest absolute reduction in the median GAD7 scores, with higher residual anxiety.

The Wilcoxon signed-rank test revealed statistically significant reductions in the GAD-7 scores between admission and discharge for participants in the emotion-focused, problem-focused, and social-support-focused coping groups (*p*-adjusted < 0.001 for all). In contrast, the change in the avoidant coping group did not reach statistical significance (*p*-adjusted = 0.29), likely due to the small sample size (n = 4). The reported *p*-adjusted values account for multiple comparisons across the coping groups, reducing the risk of Type I errors and increasing the robustness of the findings. These results support the association between engaged coping styles and meaningful reductions in anxiety symptoms during hospitalization (Table 7).

### 3.5. Correlations Between Coping Styles and Anxiety and Depression Scores at Admission and Discharge

Table 8 presents the Spearman’s rank correlation coefficients (*ρ*) and their corresponding *p*-values, illustrating the relationships between the coping styles and the GAD7/PHQ9 scores at admission and discharge. The table highlights how each coping style correlates with the anxiety (GAD7) and depression (PHQ9) levels over time. Avoidant coping showed no significant correlations with either the anxiety or depression scores at admission or discharge (*p* > 0.05), likely due to the small sample size for this group. Emotion-focused coping exhibited significant negative correlations with the GAD7 at discharge (*ρ* = −0.33, *p* < 0.01) and the PHQ9 at discharge (*ρ* = −0.25, *p* = 0.01), suggesting that higher reliance on this coping style was associated with lower anxiety and depression scores. Problem-focused coping was negatively correlated with the GAD7 at admission (*ρ* = −0.20, *p* = 0.04) and the PHQ9 at admission (*ρ* = −0.21, *p* = 0.03), indicating that greater use of this coping style was linked to lower anxiety and depression levels at the start of hospital admission. Social-support-focused coping demonstrated significant positive correlations with the GAD7 at both admission (*ρ* = 0.22, *p* = 0.03) and discharge (*ρ* = 0.24, *p* = 0.01), as well as with the PHQ9 at admission (*ρ* = 0.23, *p* = 0.02). This suggests that individuals relying on social support tend to report higher anxiety and depression scores compared to those relying on other coping styles. These results emphasize the nuanced role of coping strategies in influencing psychological outcomes. Emotion-focused coping was associated with lower anxiety and depression levels at discharge, while problem-focused coping correlated with lower scores at admission. Conversely, social-support-focused coping was linked to higher psychological distress at both time points, highlighting potential challenges in managing anxiety and depression within this group.

### 3.6. Multinomial Logistic Regression Analysis of Coping Style Predictors

The analysis of the multinomial logistic regression models was conducted to identify significant predictors of coping style selection, with the PHQ9 discharge scores as the primary independent variable and the coping styles as the response categories. This analysis aimed to evaluate how residual depressive symptoms at discharge influence the likelihood of adopting specific coping mechanisms, such as emotion-focused, problem-focused, or social-support-focused coping, compared to avoidant coping. The results revealed significant associations between the PHQ9 discharge scores and the likelihood of adopting all three coping styles compared to avoidant coping. Higher PHQ9 discharge scores were consistently associated with increased odds of using these coping strategies. For instance, the odds of adopting emotion-focused coping increased approximately threefold (OR = 3.07) for every one-unit increase in the PHQ9 discharge score. Similarly, individuals were approximately three times more likely to adopt problem-focused coping (OR = 3.11) and social-support-focused coping (OR = 3.52) compared to avoidant coping for each unit increase in the PHQ9 discharge score. The Nagelkerke R^2^ value of 0.062 indicates that the model explains approximately 6.2% of the variance in coping style selection, suggesting that the PHQ9 discharge scores play a modest but significant role in influencing coping mechanisms. While the model shows a statistically significant relationship, it also highlights the need to explore additional predictors to fully understand the factors influencing coping styles. The results are presented in Table 9.

### 3.7. Linear Regression Models Analyzing Predictors of Coping Styles

The linear regression analysis conducted for avoidant coping examined the predictive relationships between the sociodemographic factors, the psychological variables (GAD7 and PHQ9 scores), and the likelihood of adopting avoidant coping strategies. This analysis aimed to identify the specific factors, both psychological and demographic, that contribute to variations in avoidant coping behaviors. By integrating both the anxiety and depression scores alongside key sociodemographic predictors, the model provides insights into the complex interplay of psychological and personal factors in influencing avoidant coping. The regression model identified marital status as a significant predictor of avoidant coping. Being married was associated with a higher likelihood of using this coping style, with an estimated effect size of 1.74. This finding suggests that marital status plays a role in shaping avoidant coping behaviors, potentially reflecting differences in social dynamics or emotional responses among married individuals. Despite the significance of marital status, the model’s adjusted R^2^ value of 0.032 indicates that only 3.2% of the variance in avoidant coping was explained by the included predictors. This modest explanatory power highlights the need to explore additional variables, such as personality traits, coping resources, or situational factors, to fully understand the determinants of avoidant coping. Although only marital status is reported in the final model, the analysis initially included multiple variables (age, sex, residence, smoking status, education, GAD-7, and PHQ-9 scores upon admission and discharge). These variables were removed through backward elimination due to a lack of statistical significance. Marital status remained the only significant predictor of avoidant coping. The detailed results are presented in Table 10.

The linear regression analysis for social-support-focused coping evaluated the influence of the sociodemographic factors and psychological variables (PHQ9 admission scores) on the likelihood of adopting this coping strategy. This analysis aimed to understand how marital status and baseline depressive symptoms shape reliance on social support as a primary coping mechanism. By integrating these predictors, the model sheds light on the interpersonal and psychological factors that contribute to social-support-focused coping. The model identified marital status and the PHQ9 admission scores as significant predictors. Being married was strongly associated with a higher likelihood of using social-support-focused coping, with an estimated effect size of 2.58. This suggests that married individuals may have greater access to, or a preference for, social connections and interpersonal support when managing stress. Additionally, the PHQ9 admission scores were positively associated with social-support-focused coping, indicating that higher baseline levels of depressive symptoms were linked to an increased reliance on this coping strategy. This relationship highlights the role of emotional distress in driving individuals to seek social support as a coping mechanism. The adjusted R^2^ value of 0.097 indicates that approximately 9.7% of the variance in social-support-focused coping was explained by the included predictors. While the model demonstrates significant associations, additional factors, such as social network size, personality traits, or cultural influences, may further clarify the use of this coping style. The detailed results are presented in Table 11.

The linear regression analysis for emotion-focused coping evaluated the association between the PHQ9 discharge scores and the likelihood of adopting this coping style, considering its relationship with the psychological and sociodemographic variables. This analysis aimed to explore how the depressive symptom levels at discharge, alongside other factors, influence the preference for emotion-focused coping strategies.

The model identified the PHQ9 discharge scores as a significant predictor of emotion-focused coping. The negative association suggests that individuals with lower depressive symptom levels at discharge were less likely to rely on emotion-focused coping. This relationship may reflect the emotional state of individuals, as higher depressive symptoms might drive a greater focus on managing emotions as a coping mechanism.

The adjusted R^2^ value of 0.041 indicates that the PHQ9 discharge scores account for approximately 4.1% of the variance in emotion-focused coping. While the model explains a small proportion of the variance, the significant association underscores the role of depressive symptoms in influencing the use of emotion-focused strategies. The detailed results are presented in Table 12.

The linear regression analysis for problem-focused coping examined the influence of age, marital status, and other sociodemographic and psychological variables on the likelihood of adopting this coping style. This analysis aimed to understand how specific demographic factors shape problem-focused coping preferences, which emphasize actively addressing stressors. The model identified age and marital status as significant predictors of problem-focused coping. Age was positively associated with this coping style, suggesting that older individuals were more likely to rely on problem-focused strategies. This finding aligns with the idea that age may be associated with greater life experience and problem-solving capacity. Additionally, marital status was a significant predictor, with married individuals being more likely to adopt problem-focused coping. This could reflect the potential for married individuals to leverage shared resources, support, or collaborative problem-solving with their partners. The adjusted R2 value of 0.049 indicates that age and marital status explain approximately 4.9% of the variance in problem-focused coping. While the model’s explanatory power is modest, the significant predictors highlight the relevance of demographic factors in shaping coping styles. These results underscore the role of age and marital status in influencing problem-focused coping preferences, providing valuable insights into how demographic characteristics contribute to active stress management strategies. The detailed results are presented in Table 13.

## 4. Discussion

This study explored the relationship between coping styles, anxiety, depression, and sociodemographic factors in tuberculosis patients, identifying key predictors of psychological outcomes. The results revealed significant differences in mental health across coping styles, with problem-focused coping being associated with better psychological recovery, while social-support-focused coping was linked to higher residual distress.

Patients experienced a significant reduction in anxiety and depression during hospitalization. Despite this improvement, the coping styles influenced the residual distress at discharge. Patients relying on social-support-focused coping showed the largest reduction in the PHQ9 scores (from 13 to 4), yet they still had higher residual distress compared to the other groups. This suggests that while social support can be beneficial, it may not fully mitigate psychological distress, possibly because it fosters emotional dependence rather than promoting individual resilience. Another possible explanation is the variability in the quality or availability of support systems. In some cases, individuals may rely on social support but receive inadequate emotional or practical help, leading to unmet needs and prolonged distress. Additionally, reliance on external validation for emotional regulation may limit the development of internal coping resources. Cultural norms could also shape expectations around communal support and influence how distress is expressed and processed, potentially making some individuals more vulnerable when such support is insufficient or misaligned with their needs. In contrast, avoidant coping, though often considered maladaptive, was associated with the lowest final anxiety and depression scores (PHQ9 = 0.5, GAD7 = 0), indicating that, for some individuals, psychological distancing from stressors may serve as a short-term adaptive strategy.

While our findings indicate a significant reduction in anxiety and depression symptoms during hospitalization, it is important to note that this improvement cannot be solely attributed to anti-tuberculosis treatment. This study was not designed to isolate the effects of treatment on the psychological outcomes, and we did not control for other potential influences such as the psychological impact of hospitalization itself, patient–clinician interactions, or natural adaptation over time. As such, while TB treatment may contribute to psychological improvement, we cannot conclude this definitively based on our current data. Further studies using controlled or longitudinal designs are needed to explore the specific contribution of treatment to mental health changes in TB patients.

Age was a significant factor in determining the coping styles and their effectiveness. Older patients (median = 48 years) were more likely to use problem-focused coping, potentially due to their greater life experience and problem-solving skills, while younger patients (median = 39 years) adopted social-support-focused or emotion-focused coping. The Kruskal–Wallis test confirmed that older individuals preferred strategies that directly addressed stressors, and the regression analyses showed that age was a significant predictor of problem-focused coping (β = 0.08, *p* = 0.008), further reinforcing the idea that problem-solving tendencies increase with age.

Marital status also emerged as an important determinant of coping style. Married patients were significantly more likely to adopt problem-focused coping, which was associated with better psychological outcomes. This can be explained by the emotional and practical support provided by a partner, which may facilitate better problem-solving and stress management. The multinomial regression further supported this finding, showing that married individuals had higher odds of using problem-focused coping, while unmarried individuals were more likely to rely on emotion-focused coping, potentially as a self-regulatory mechanism in the absence of strong social support. This aligns with previous findings indicating that social support enhances coping but does not necessarily eliminate psychological distress, particularly when it fosters dependence rather than proactive problem-solving [5].

Other sociodemographic factors, such as education and smoking behavior, also influenced coping preferences. A higher education level was slightly associated with problem-focused coping, though this relationship was not statistically significant. The smoking status showed an interesting trend, with smokers more likely to engage in avoidant coping (7%) compared to non-smokers (0%), suggesting that individuals who engage in avoidant behaviors in response to stress may also turn to smoking as a maladaptive coping mechanism. While this trend was not statistically significant, it highlights an area for future research on the intersection between behavioral health and psychological coping in TB patients.

The study by Ni Putu Wulan Purnama Sari and colleagues explored the stress levels and coping strategies among newly diagnosed tuberculosis patients, comparing the intensive and advanced phases of treatment. They found no significant differences in the stress levels or coping strategies between the two phases but identified emotional reactions such as anger, loss of control, and nervousness as key areas requiring attention during the intensive treatment phase. In comparison, our study focuses on the psychological distress experienced by TB patients and how different coping styles are associated with variations in the anxiety and depression levels at admission and discharge. Unlike the findings of Sari et al., our results highlight significant variability in the psychological outcomes based on the coping style. For instance, social-support-focused coping was linked to persistently higher levels of distress, while problem-focused coping was associated with better outcomes [11].

Rajeev and Pradeep’s study highlights the prevalence of severe depression in 45.7% of TB patients, with significant associations between depression and factors such as perceived stigma, comorbid conditions like diabetes, and sociodemographic variables such as education and marital status. These findings align with our study, which also identified high levels of anxiety and depression among TB patients, particularly at admission. However, while Rajeev and Pradeep focused on the distribution of coping strategies among patients, our study extends this understanding by directly linking the coping styles to psychological outcomes, such as reductions in the GAD7 and PHQ9 scores at discharge. For example, our findings reveal that avoidant coping, while often considered maladaptive, was associated with the lowest residual psychological distress, albeit in a small subgroup of patients. This complements Rajeev and Pradeep’s observation that emotion-focused coping was the most common strategy among TB patients but highlights the nuanced effectiveness of different coping styles in managing mental health challenges [12].

In Mason et al.’s study, a significant proportion of patients (57%) employed emotion-focused coping, a finding that aligns with our study, where emotion-focused coping was the most prevalent style (44%). However, while Mason et al. observed that emotion-focused coping was associated with moderate reductions in distress, our results indicate that this style was moderately effective, with the PHQ9 scores dropping from 9 to 2 and the GAD7 scores from 7 to 2 for this group. Conversely, problem-focused coping in our study demonstrated stronger psychological benefits, with patients showing a larger reduction in anxiety and depression. Furthermore, Mason et al. found that avoidant coping was the least effective in reducing distress. In contrast, our study revealed unexpectedly low residual distress in the avoidant coping group (PHQ9 scores dropping from 4.5 to 0.5 and GAD7 scores from 3 to 0), though this result must be interpreted cautiously due to the small sample size (n = 4). These differences underscore the importance of cultural and contextual factors influencing the effectiveness of coping mechanisms [17].

The findings of Doherty et al. align with our study’s observation of significant rates of anxiety and depression in TB patients, as assessed using the GAD-7 and PHQ-9 scales. Their review noted mental health conditions in up to 70% of TB patients, a statistic that underscores the importance of screening for psychiatric comorbidities in this population. In contrast, our study used the coping styles as a lens to investigate how patients psychologically adapt to their TB diagnosis and its management. Doherty et al. also emphasized the bidirectional relationship between TB and mental health, where TB treatments can exacerbate psychiatric symptoms due to adverse drug effects. Although our study did not focus on adverse effects of treatment, we observed that different coping styles correlated with varied levels of anxiety and depression improvement, suggesting that psychological resilience and adaptive coping mechanisms could mitigate these challenges [23].

The findings of this study have significant implications for integrating mental health support into tuberculosis care. The strong association between coping styles and psychological outcomes, as demonstrated by the PHQ9 and GAD7 scores, underscores the need for tailored interventions that promote adaptive coping strategies. For instance, patients who relied on problem-focused coping experienced better psychological outcomes, suggesting that structured interventions, such as problem-solving therapy or cognitive–behavioral techniques, could further enhance resilience and reduce anxiety and depression. Conversely, the persistently high residual distress observed in patients using social-support-focused coping highlights the importance of supplementing social support with individualized therapeutic approaches, such as emotional regulation training or mindfulness-based stress reduction. Additionally, the significant influence of sociodemographic factors, such as age, marital status, and educational attainment, suggests that psychological care should be personalized to address these contextual factors. The fact that age and marital status were strong predictors of the coping styles suggests that psychological interventions should be tailored to these characteristics. While marital status emerged as a statistically significant predictor of avoidant coping, the model’s adjusted R^2^ was 0.032, indicating that only 3.2% of the variance in avoidant coping was explained. This low explanatory power suggests that other factors may influence the use of avoidant strategies. Future studies should consider including psychological traits (e.g., personality dimensions), perceived coping resources, or contextual stressors to improve the model performance and gain deeper insights into the determinants of avoidant coping behaviors. Younger and unmarried patients may benefit more from structured problem-solving interventions to develop adaptive coping skills, while older and married patients could leverage existing resources to enhance their problem-focused approaches. Furthermore, integrating mental health screening tools (GAD7, PHQ9) into routine TB care could help identify patients at risk of persistent distress, ensuring that psychological support is adapted to their coping tendencies. 

This study provides a comprehensive analysis of the psychological impact of tuberculosis by integrating validated tools such as the COPE questionnaire, GAD7, and PHQ9 scales to evaluate patients’ coping mechanisms, anxiety, and depression. A key strength lies in the robust statistical approach, including the Wilcoxon signed-rank test, multinomial logistic regression, and linear regression models, which allowed for nuanced insights into the relationships between coping styles, psychological outcomes, and sociodemographic factors. Unlike many studies that focus solely on the prevalence of psychological distress, this study explored how specific coping mechanisms influence mental health, revealing that problem-focused coping is associated with better outcomes, while social-support-focused coping is linked to higher residual distress. Additionally, this study’s focus on a well-defined cohort of tuberculosis patients from a clinical setting ensures the relevance of the findings to TB care. The inclusion of sociodemographic variables adds another layer of depth, highlighting important predictors of coping behavior, such as age, marital status, and education level.

These findings align with the broader research on coping strategies in chronic illnesses. For instance, Cheng et al. (2019) [29] conducted a meta-ethnography exploring patients’ experiences with multiple chronic conditions, revealing that coping mechanisms are deeply influenced by individual appraisals, efforts to maintain normalcy, and the social context. This underscores the necessity of considering the quality and availability of support systems, as well as cultural norms, when evaluating the effectiveness of social-support-focused coping strategies [29].

Despite its strengths, this study has several limitations that should be acknowledged. The sample size, particularly in the avoidant coping group (n = 4), was small, which limits the generalizability of findings for this subgroup. As such, the conclusions related to avoidant coping should be regarded as exploratory and interpreted with caution, warranting further investigation in larger, more diverse cohorts. Moreover, this study was conducted in a single-center setting, which may limit the generalizability of the findings to other populations or healthcare systems. Future research would benefit from a multicenter design to improve the sample diversity and external validity. Additionally, this study relied on self-reported scales, which are subject to response bias, including social desirability bias. Moreover, the requirement that participants fully understand the study procedures may have introduced a selection bias, potentially excluding individuals with lower literacy, cognitive impairments, or other vulnerabilities. These populations may be at higher risk of maladaptive coping and psychological distress, and their underrepresentation could limit the generalizability of our findings. Future studies may consider the use of trained facilitators or assisted questionnaire completion to improve the inclusivity. Another limitation is the lack of longitudinal follow-up beyond discharge, which would have provided insights into the long-term psychological outcomes of TB patients and the sustainability of coping mechanisms. This study also did not explore cultural or contextual factors, such as religious or spiritual coping, which may play a significant role in how patients manage stress in certain settings. Finally, while the analysis linked coping styles to psychological outcomes, it did not account for potential confounding factors, such as comorbidities or the side effects of TB treatment, which might also influence mental health. Addressing these limitations in future research would strengthen the evidence base and provide a more comprehensive understanding of the interplay between TB, coping, and mental health. In particular, longitudinal study designs would help track the evolution of coping styles and psychological symptoms beyond hospitalization, offering insight into whether certain coping mechanisms—such as social-support-focused coping—remain beneficial or become maladaptive over time. Additionally, experimental designs testing tailored psychological interventions, such as problem-solving therapy or mindfulness training, could determine their effectiveness in promoting adaptive coping styles and improving mental health. Investigating the interplay between mental health and treatment adherence in multidrug-resistant TB (MDR-TB) patients is another critical area, as these patients often experience higher levels of psychological distress. Lastly, exploring the integration of mental health screening tools like the GAD7 and PHQ9 into routine TB care could provide evidence of their utility in improving early detection and intervention for at-risk patients.

## 5. Conclusions

This study provides valuable insights into the psychological impact of tuberculosis and the role of coping mechanisms in influencing mental health outcomes. Using validated scales (GAD7, PHQ9, and COPE), we demonstrated significant reductions in anxiety and depression during hospitalization, although residual psychological distress persisted in patients relying on certain coping styles, such as social-support-focused coping. Problem-focused coping was associated with better psychological outcomes, while avoidant coping, though less prevalent, showed the lowest residual anxiety and depression scores. Sociodemographic factors, such as age, marital status, and education, play a critical role in shaping coping behaviors, underscoring the need for tailored psychological interventions.

Our findings emphasize the importance of integrating mental health care into tuberculosis management. Routine screening for anxiety and depression using tools like the GAD7 and PHQ9, combined with interventions that promote adaptive coping strategies, may enhance psychological resilience and indirectly support treatment outcomes in TB patients. As this study did not directly assess adherence, future research should evaluate this relationship using validated adherence instruments (e.g., Morisky Medication Adherence Scale).

## Figures and Tables

**Figure 1 healthcare-13-01042-f001:**
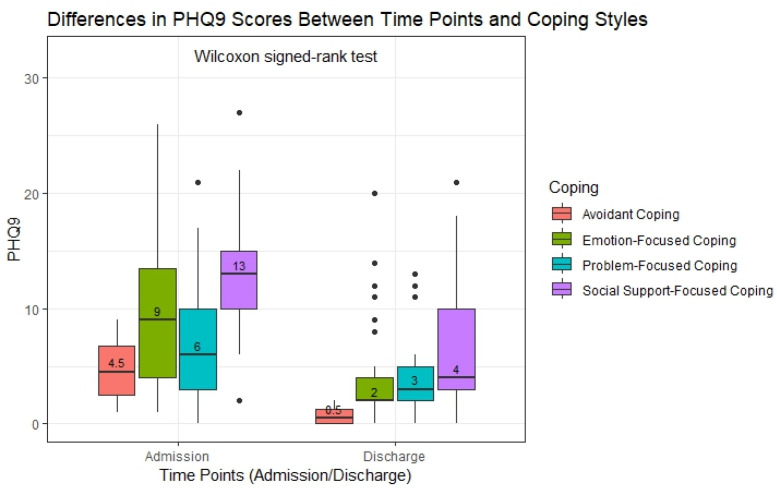
Differences in the PHQ9 scores across time points and coping styles (Wilcoxon signed-rank test). Abbreviations: PHQ9—Patient Health Questionnaire-9.

**Figure 2 healthcare-13-01042-f002:**
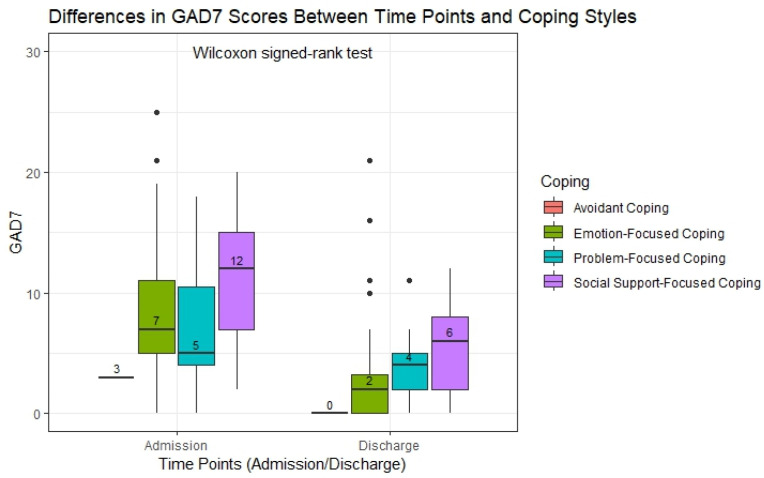
Differences in the GAD7 scores across time points and coping styles (Wilcoxon signed-rank test). Abbreviations: GAD7—Generalized Anxiety Disorder 7-item.

**Table 1 healthcare-13-01042-t001:** Summary of the numerical variables and statistical testing results of the Shapiro–Wilk normality test.

Variable	Median (Q25–Q75)	*p*-Value
Age	43.50 (30.00–53.25)	<0.001 *
GAD7_Admission	7.00 (4.00–12.00)	<0.001 *
GAD7_Discharge	2.00 (1.00–5.00)	<0.001 *
PHQ9_Admission	9.00 (4.00–13.00)	<0.001 *
PHQ9_Discharge	3.00 (2.00–5.00)	<0.001 *
Avoidant Coping	28.00 (25.00–30.25)	<0.001 *
Social-Support-Focused Coping	31.00 (28.00–32.00)	<0.001 *
	**Mean ± Standard Deviation**	
Problem-Focused Coping	32.72 ± 4.08	0.158
Emotion-Focused Coping	34.7 ± 3.82	0.242

Abbreviations: GAD7—Generalized Anxiety Disorder 7-item; PHQ9—Patient Health Questionnaire-9; Q25–Q75—interquartile range, *p*-value—Shapiro–Wilk test result. * It indicates statistically significant results.

**Table 2 healthcare-13-01042-t002:** Distribution of categorical variables.

Variable	Group	Count (Proportion)
Sex	Female	43 (43%)
	Male	57 (57%)
Residence	Rural	49 (49%)
	Urban	51 (51%)
Smoking Status	Smoker	57 (57%)
	Non-Smoker	43 (43%)
Educational Level	High School or Less	67 (67%)
	Higher Education	33 (33%)
Marital Status	Married	66 (66%)
	Single	34 (34%)
Coping Style	Avoidant Coping	4 (4%)
	Emotion-Focused Coping	44 (44%)
	Problem-Focused Coping	31 (31%)
	Social-Support-Focused Coping	21 (21%)

**Table 3 healthcare-13-01042-t003:** Differences in coping styles across categorical sociodemographic variables (Chi-squared Pearson test).

Variable	Group	Avoidant Coping	Emotion-Focused Coping	Problem-Focused Coping	Social Support-Focused Coping	*p*-Value
Sex	Female	7%	42%	35%	16%	*p* = 0.40
	Male	2%	46%	28%	25%	
Residence	Rural	4%	41%	31%	24%	*p* = 0.85
	Urban	4%	47%	31%	18%	
Smoking Status	Smoker	7%	44%	23%	26%	*p* = 0.05 *
	Non-Smoker	0%	44%	42%	14%	
Educational Level	High School or Less	3%	45%	33%	19%	*p* = 0.80
	Higher Education	6%	42%	27%	24%	
Marital Status	Married	5%	48%	29%	18%	*p* = 0.56
	Single	3%	35%	35%	26%	

* It indicates statistically significant results.

**Table 4 healthcare-13-01042-t004:** Differences in coping styles across numerical sociodemographic variables (Kruskal–Wallis test).

Coping Style	Age (Median)	*p*-Value
Avoidant Coping	41.50	*p* = 0.24
Emotion-Focused Coping	41.50	
Problem-Focused Coping	48.00	
Social-Support-Focused Coping	39.00	

**Table 5 healthcare-13-01042-t005:** Differences in the GAD7 and PHQ9 scores at admission and discharge across coping styles (Kruskal–Wallis test).

Variable		Avoidant Coping	Emotion-Focused Coping	Problem-Focused Coping	Social-Support-Focused Coping	*p*-Value
GAD7	Admission	3.00	7.00	5.00	12.00	<0.01 *
	Discharge	0.00	2.00	4.00	6.00	<0.001 *
PHQ9	Admission	4.50	9.00	6.00	13.00	<0.01 *
	Discharge	0.50	2.00	3.00	4.00	<0.01 *

Abbreviations: GAD7—Generalized Anxiety Disorder 7-item; PHQ9—Patient Health Questionnaire 9. * It indicates statistically significant results.

**Table 6 healthcare-13-01042-t006:** Differences in the PHQ9 scores across time points and coping styles (Wilcoxon signed-rank test).

Coping Style	Median Score at Admission	Median Score at Discharge	*p*-Adjusted Value
Avoidant Coping	4.5	0.5	*p* = 0.5
Emotion-Focused Coping	9	2	<0.001 *
Problem-Focused Coping	6	3	<0.001 *
Social-Support-Focused Coping	13	4	<0.001 *

* It indicates statistically significant results.

**Table 7 healthcare-13-01042-t007:** Differences in the GAD7 scores across time points and coping styles (Wilcoxon signed-rank test).

Coping Style	Median Score at Admission	Median Score at Discharge	*p*-Adjusted Value
Avoidant Coping	3	0	*p* = 0.29
Emotion-Focused Coping	7	2	<0.001 *
Problem-Focused Coping	5	4	<0.001 *
Social-Support-Focused Coping	12	6	<0.001 *

* It indicates statistically significant results.

**Table 8 healthcare-13-01042-t008:** Spearman’s rank correlations between the coping styles and the GAD7 and PHQ9 Scores at admission and discharge.

	GAD7_Admission		GAD7_Discharge		PHQ9_Admission		PHQ9_Discharge	
Coping Style	ρ (Rho)	*p*-Value	ρ (Rho)	*p*-Value	ρ (Rho)	*p*-Value	ρ (Rho)	*p*-Value
Avoidant Coping	0.11	0.29	0.09	0.37	0.15	0.13	0.05	0.65
Emotion-Focused Coping	−0.11	0.26	−0.33	<0.01 *	−0.18	0.08	−0.25	0.01 *
Problem-Focused Coping	−0.20	0.04 *	0.03	0.80	−0.21	0.03 *	−0.07	0.50
Social-Support-Focused Coping	0.22	0.03 *	0.24	0.01 *	0.23	0.02 *	0.09	0.35

Abbreviations: GAD7—Generalized Anxiety Disorder 7-item; PHQ9—Patient Health Questionnaire-9; *ρ*—Spearman’s rank correlation coefficient, *p*-value—Spearman’s rank correlation test result. * It indicates statistically significant results.

**Table 9 healthcare-13-01042-t009:** Multinomial logistic regression predicting coping styles based on the PHQ9 scores at discharge.

Predictors	Odds Ratios	CI	*p*-Value	Response
PHQ9_Discharge	3.07	1.04–9.12	0.043 *	Emotion-Focused Coping
PHQ9_Discharge	3.11	1.05–9.24	0.041 *	Problem-Focused Coping
PHQ9_Discharge	3.52	1.18–10.47	0.024 *	Social-Support-Focused Coping
Nagelkerke R^2^ = 0.062				

Abbreviations: CI—confidence interval, *p*-value—Wald test result. * It indicates statistically significant results.

**Table 10 healthcare-13-01042-t010:** Linear regression model for avoidant coping.

Predictors	Estimates	CI	*p*-Value
Marital Status	1.74	0.07–3.42	0.042 *
R^2^ adjusted = 0.032			

Abbreviations: CI—confidence interval, *p*-value—Student’s *t*-test result. * It indicates statistically significant results.

**Table 11 healthcare-13-01042-t011:** Linear regression model for social-support-focused coping.

Predictors	Estimates	CI	*p*-Value
Marital Status	2.58	0.70–4.46	0.008 *
PHQ9 Admission	0.19	0.05–0.33	0.010 *
R^2^ adjusted = 0.097			

Abbreviations: CI—confidence interval, *p*-value—Student’s *t*-test result. * It indicates statistically significant results.

**Table 12 healthcare-13-01042-t012:** Linear regression model for emotion-focused coping.

Predictors	Estimates	CI	*p*-Value
PHQ9 Discharge	−0.20	−0.38–−0.03	0.025 *
R^2^ adjusted = 0.041			

Abbreviations: CI—confidence interval, *p*-value—Student’s *t*-test result. * It indicates statistically significant results.

**Table 13 healthcare-13-01042-t013:** Linear regression model for problem-focused coping.

Predictors	Estimates	CI	*p*-Value
Age	0.08	0.01–0.14	0.008 *
Marital Status	1.91	0.11–3.71	0.037 *
R^2^ adjusted = 0.049			

Abbreviations: CI—confidence interval, *p*-value—Student’s *t*-test result. * It indicates statistically significant results.

## Data Availability

The data presented in this study are available on request from the corresponding author. The data are not publicly available due to reasons concerning the privacy of the subjects.

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
