# Peer review of "The Impact of Maladaptive Coping Styles on Psychological Outcomes in Tuberculosis Patients"

_healthcare, 2025, doi:10.3390/healthcare13091042_

Round 1

Reviewer 1 Report

Comments and Suggestions for Authors

This study explores a crucial yet frequently neglected aspect of tuberculosis (TB) care—how patients mentally manage their diagnosis and treatment. The connection between physical health and mental well-being is becoming more recognized, and this research highlights the role that various coping mechanisms play in psychological distress among TB patients. Below is an in-depth evaluation of the study’s strengths, potential areas for improvement, and suggestions to further enhance its significance.

The study seeks to answer an important question: *How do different coping styles impact the psychological well-being of tuberculosis patients?* Specifically, it examines how problem-focused, emotion-focused, social support-focused, and avoidant coping strategies influence levels of anxiety and depression in TB patients, using validated psychological scales (GAD-7 for anxiety and PHQ-9 for depression).

This is a particularly valuable inquiry because TB is not just a physical illness; it comes with significant psychosocial burdens, including stigma, isolation, and economic hardship. By investigating how patients manage this distress, the study contributes to a more holistic understanding of TB care, emphasizing the need for mental health support alongside medical treatment.

While the medical aspects of TB—diagnosis, treatment, and prevention—have been extensively studied, the psychological burden of the disease remains underexplored. The study highlights that TB patients frequently experience anxiety and depression, yet their coping mechanisms are rarely considered when developing treatment plans.

TB disproportionately affects populations in low- and middle-income countries, where access to mental health services is limited. The stigma surrounding TB can compound psychological distress, making it harder for patients to seek social support. Coping strategies are not one-size-fits-all; understanding which mechanisms are most effective could lead to more personalized mental health interventions.

By focusing on how coping styles influence mental health outcomes, this study addresses an important gap in TB research. However, the study would be even stronger if it expanded on why certain coping styles are more common in TB patients and how healthcare providers can support healthier coping mechanisms.

Compared to previous studies on mental health in TB patients, this research provides a more structured and quantitative approach to understanding coping mechanisms. It offers several important contributions:

- It categorizes coping mechanisms into problem-focused, emotion-focused, social support-focused, and avoidant coping, providing a more nuanced understanding of how patients react to psychological distress.

- It uses validated psychological assessment tools (COPE, GAD-7, PHQ-9) to measure mental health outcomes, lending credibility to the findings.

- The inclusion of socio-demographic factors (age, marital status, education, smoking status) allows for a deeper exploration of what influences coping styles.

This study does a great job of demonstrating that not all coping strategies are equally beneficial—for example, social support-focused coping, while seemingly positive, was associated with higher residual distress after hospitalization. This challenges common assumptions and suggests that relying too much on external support may not always be the best approach.

However, one potential area for improvement is in contextualizing these findings. Why do some TB patients lean toward one coping style over another? Are these coping styles influenced by cultural, economic, or healthcare system factors? A qualitative component (such as patient interviews) could enrich the study by providing a human perspective on the statistics.

While the study is well-designed, there are a few areas where the methodology could be strengthened:

a) Sample Size & Representation

  1. The avoidant coping group had an extremely small sample size (n=4), which makes it difficult to draw meaningful conclusions about this subgroup. A larger and more diverse sample would provide more robust and generalizable findings.
  2. The study would benefit from a multicenter approach (recruiting patients from multiple hospitals or regions) to ensure that findings are not specific to one particular setting.

b) Study Design & Follow-Up

  1. Since this is a cross-sectional study, it only captures data at one point in time (during hospitalization). However, coping mechanisms and mental health outcomes can change over time. A longitudinal design (following patients after discharge) would provide valuable insights into the long-term effects of different coping styles.
  2.  For example, does social support-focused coping continue to be linked to distress after treatment, or does its impact fade over time? Answering this would help refine psychological interventions.

c) Controlling for Confounding Variables

  1. The study does not fully account for potential confounders
  2. Pre-existing mental health conditions (e.g., a patient with prior depression may naturally cope differently).
  3. The impact of TB treatment side effects (e.g., medication-induced depression).
  4. Economic status and healthcare access, which could influence both coping strategies and mental health outcomes.

Using multivariate regression models that control for these factors would help ensure that the observed effects are truly due to coping styles and not external variables.

Overall, the conclusions are consistent with the findings, but some claims could be better contextualized and nuanced.

  1. The study correctly identifies problem-focused coping as the most beneficial strategy, which aligns with existing psychological research.
  2. However, the avoidant coping findings need to be interpreted with caution. While this group showed the lowest anxiety and depression scores post-treatment, the extremely small sample size (n = 4) means this result may not be reliable. The conclusion that avoidance may be an adaptive short-term strategy should be framed as a hypothesis rather than a definitive statement.
  3. The study suggests that social support-focused coping may not always be beneficial, as patients in this category still reported residual distress. While this is an important and unexpected finding, the study could explore why this is the case. Is it because these individuals rely on external validation for emotional well-being? Or is it that their support systems are not equipped to help with TB-related stress?

The discussion would benefit from a clearer acknowledgment of the study’s limitations, as well as a call for future research to confirm and expand upon these findings.

The references are relevant and include both foundational psychological theories and recent studies on TB and mental health. However, incorporating broader psychiatric research on coping strategies in chronic disease patients could provide additional theoretical depth.

The tables present the data clearly, but they could be more reader-friendly by highlighting the most significant findings (e.g., bolding statistically significant results).

Some figures lack detailed legends, making it harder to interpret key takeaways. Adding brief descriptions of what each table/figure represents would improve clarity.

A graphical summary (e.g., a flowchart showing how different coping styles influence mental health) could enhance readability and accessibility for non-specialist readers.

Final Recommendations for Improvement

  1. Expand the sample size, particularly for the avoidant coping group.
  2. Consider a longitudinal design to track coping strategies and mental health over time.
  3. Control for additional confounders, such as pre-existing mental health conditions and TB medication side effects.
  4. Provide more contexts on why certain coping strategies are preferred by TB patients.
  5. Clarify conclusions, especially regarding avoidant coping, to avoid overgeneralization.
  6. Enhance tables and figures with clearer explanations and visual summaries.

By implementing these changes, the study could make an even stronger contribution to understanding the mental health challenges faced by TB patients and how to best support them in their recovery.

Reviewer 2 Report

Comments and Suggestions for Authors

Tuberculosis remains the world's leading health problem. The peer-reviewed article is devoted to the identification of factors that influence the adherence to treatment in patients with tuberculosis (TB). Undoubtedly, this is a very important task for public health.

The introduction is excessively lengthy and detailed for an article; it resembles a chapter of a review. It would be advisable to shorten and refine it, emphasizing the most salient points for specific practical work.

Major

A number of questions have been raised concerning the formation of groups, thereby raising concerns about the objective assessment of the identification of psychological distress causes:

  • Were the patients previously treated or newly diagnosed with TB? The patient's status, whether previously treated or newly diagnosed, can have a strong influence on their psychological well-being.
  • What anti-tuberculosis chemotherapy have patients had? The psychological state of patients may be affected by almost all anti-TB drugs, including the first-line drugs (cycloserine, linezolid, isoniazid, levofloxacin, etc.). These include sleep disturbances to acute psychosis, color blindness, depressive syndrome, etc. Different combinations of TB drugs can have different side effects. These side effects are quite common.
  • Were the patients being treated with antidepressants or other psychotropic medications?

Reviewer 3 Report

Comments and Suggestions for Authors

The Impact of Maladaptive Coping Styles on Psychological Outcomes in Tuberculosis Patients

Dear author and editor:

This article talked about effect of coping style on psychological features of TB patients. The article could be published after a minor revision and I have some comments on it.

  • Does treatment ameliorate this psychological outcome?
  • Is there any relation between antidepressant and the anti-bacterial response?
  • These psychological outcomes related just to patient with active or latent tuberculosis.
  • Are there any references about the specific mechanism of the relation between the mental health problem and tuberculosis?

                                     Thank you very much, best regards

Reviewer 4 Report

Comments and Suggestions for Authors

Here is a suggested comment to the authors regarding the potential selection bias in their study: One concern regarding the inclusion criteria is the requirement that all participants must "fully understand the study procedures" to ensure reliable responses to psychological assessments. This criterion may introduce selection bias, as it could systematically exclude individuals with lower cognitive ability, lower literacy levels, or other vulnerabilities. Given that maladaptive coping styles may be more prevalent among such individuals, this exclusion may limit the generalizability of the findings.

To address this, I suggest that the authors:

 • Acknowledge this potential bias explicitly in the limitations section of the manuscript.

• Clarify how participant understanding was assessed—for example, whether any standardized tool or simplified explanation was used.

• Consider alternative approaches to minimize exclusion, such as providing additional assistance (e.g., trained facilitators or simplified explanations) to ensure broader participation.

The manuscript states that the Friedman test was used to analyze differences in GAD-7 and PHQ-9 scores at admission and discharge. However, the Friedman test is typically appropriate for comparing three or more related measurements within the same group. If only two time points (admission and discharge) were analyzed, the Wilcoxon signed-rank test (for non-normally distributed data) or the paired t-test (for normally distributed data) would be more appropriate.

To improve the clarity and accuracy of the statistical analysis, I recommend:

• Clarifying the number of time points analyzed in the study. If only two time points were included, the Friedman test should be replaced with the Wilcoxon signed-rank test or the paired t-test.

• Justifying the choice of the Friedman test if more than two time points were analyzed.

• Ensuring that data distribution is assessed before choosing a parametric or non-parametric test.

In the regression model analyzing predictors of avoidant coping, only marital status is reported as a significant predictor. However, it is unclear whether other variables were included in the model but found to be non-significant, or if marital status was the only predictor tested.

To improve clarity

Explicitly listing all variables that were included in the regression model, even if they were non-significant. If other predictors (e.g., sociodemographic factors, GAD-7, PHQ-9) were tested but not significant, they should either be presented in Table 8 or at least acknowledged in the

Discussing the limitation of the model’s low explanatory power (adjusted R2R^2R2 = 0.032) and suggesting potential additional predictors (e.g., personality traits, coping resources) that might better explain avoidant coping.

The manuscript suggests that integrating mental health screening (using PHQ-9 and GAD-7) into TB care may enhance treatment adherence. However, PHQ-9 and GAD-7 assess depression and anxiety, respectively, and do not directly measure adherence. While mental health factors may influence adherence, a direct assessment tool (e.g., Morisky Medication Adherence Scale) would be needed to evaluate this outcome reliably. I recommend to change the conclusion in abstract.

Round 2

Reviewer 2 Report

Comments and Suggestions for Authors

I just have one more question. If the trial included newly diagnosed TB patients before the start of treatment, what kind of treatment adherence can we talk about?

Author Response

I just have one more question. If the trial included newly diagnosed TB patients before the start of treatment, what kind of treatment adherence can we talk about?

Response: Thank you for your follow-up question. We fully agree that, given the study design, treatment adherence could not be directly assessed. Our reference to adherence in the manuscript is strictly hypothetical and framed as a future research direction. As clarified in both the Abstract and Conclusion sections, we note that adherence was not measured in this study and suggest that future investigations use validated tools to explore the potential relationship between coping styles, psychological outcomes, and adherence to TB treatment.